# Identification and characterization of a membrane receptor that binds to human STC1

Hin Ting Wan, Alice HM Ng, Wang Ka Lee, Feng Shi, Chris Kong-Chu Wong

**Stanniocalcin-1 (STC1) is a hypocalcemic hormone originally identified in bony fishes. The mammalian homolog is found to be involved in inflammation and carcinogenesis, among other physiological functions. In this study, we used the TriCEPS-based ligand–receptor methodology to identify the putative binding proteins of human STC1 (hSTC1) in the human leukemia monocytic cell line, ThP-1. LC–MS/MS analysis of peptides from shortlisted hSTC1-binding proteins detected 32 peptides that belong to IGF2/MPRI. Surface plasmon resonance assay demonstrated that hSTC1 binds to immobilized IGF2R/MPRI with high affinity (10–20 nM) and capacity (Rmax 70–100%). The receptor binding data are comparable with those of (CREG) cellular repressor of E1A-stimulated gene a known ligand of IGF2R/MPRI, with Rmax of 75–80% and affinity values of 1–2 nM. The surface plasmon resonance competitive assays showed CREG competed with hSTC1 in binding to IGF2R/MPRI. The biological effects of hSTC1 on ThP-1 cells were demonstrated via IGF2R/MPRI to significantly reduce secreted levels of IL-1$\beta$. This is the first study to reveal the high-affinity binding of hSTC1 to the membrane receptor IGF2R/MPRI.**

## Introduction

The hypocalcemic polypeptide hormone stanniocalcin-1 (STC1) is secreted by a unique endocrine gland, the corpuscle of Stannius (CS) (Stannius, 1839). STC1 was released in an endocrine manner, in response to increased plasma Ca$^{2+}$, to inhibit gill and intestinal Ca$^{2+}$ uptake and renal phosphate excretion (Ellis & Wagner, 1995; Radman et al, 2002; Greenwood et al, 2009). Because mammals lack CS glands, it was assumed that the STC1 gene had been lost in evolution. Nevertheless, the mammalian homolog of STC1 was discovered in the late 90 s, when mouse and human STC cDNAs were cloned (Chang et al, 1995, 1996; Olsen et al, 1996). An interesting observation is that the mammalian STC1 protein is widely expressed (Chang et al, 1996; Varghese et al, 1998) but is barely detectable in blood (De Niu et al, 2000). In mammals, the protein was reported to be involved in pleiotropic functions (growth & development, metabolism, reproduction, and cancer) (Yeung et al,

2012). These observations suggest changes in tissue expression patterns and the mode of action of mammalian STC1. To investigate the functions of the mammalian homolog, two STC1-overexpressing transgenic mouse lines were generated (Filvaroff et al, 2002; Varghese et al, 2002). The transgenic animals exhibited dwarf phenotypes (Johnston et al, 2010). In STC1 loss-of-function mice, however, there were no observable effects on body growth and development (Chang et al, 2005). In other experiments, STC1 was shown to regulate Ca$^{2+}$ and Pi transport (Wagner et al, 1997; Madsen et al, 1998) and renal water homeostasis in rats (Turner et al, 2011). Studies have shown that STC1 inhibited transmembrane calcium currents via L-channels in cardiomyocytes (Sheikh-Hamad et al, 2003), as well as steroidogenesis (Paciga et al, 2003), mitochondrial function (Ellard et al, 2007; Pan et al, 2015), inflammation (Huang et al, 2009; Prockop & Oh, 2012), and carcinogenesis (Chang et al, 2003).

STC1 has been reported to have biological effects in fishes and mammals, but the identity of its receptor is unknown. Evidence suggests, however, that the receptor exists. Fish primary cell culture models treated with STC1 increased cytosolic cAMP levels in gill (Gu et al, 2015) and renal tubular cells (Lu et al, 1994). A study of the localization of STC1 mRNA and protein found that STC1 protein was sequestered by cells that did not express the gene (Haddad et al, 1996; Varghese et al, 1998; Deol et al, 2000; Stasko et al, 2001; Stasko et al, 2001). STC1 mRNA was limited to theca-interstitial cells in the rat ovary, whereas STC1 protein accumulated in oocytes and corpus luteum (Varghese et al, 1998). The observation of protein sequestration is not unique to STC1, there have been other reports of cellular sequestration of secreted proteins, such as in the case of corticosteroid-binding globulin in embryonic and fetal mice (Scrocchi et al, 1993) or basic fibroblast growth factor in rat spinal neurons (Blottner et al, 1997). Intriguingly, further studies using receptor-binding and in situ ligand binding analysis revealed the presence of high-affinity STC1 receptors in nephrons and hepatocyte mitochondria (McCudden et al, 2002) as well as cholesterol/lipid storage droplets of luteal cells (Paciga et al, 2003). The observation has suggested a receptor-mediated process for sequestering the protein. In recent studies, it has been demonstrated that endosome-to-Golgi and Rab32-mediated retrograde transport of STC1 to mitochondria is dependent on LRP2/megalin binding (Li et al, 2018, 2022). The studies shed light on how STC1 regulates mitochondrial metabolism. In this study, we used the TriCEPS-based

Department of Biology, Croucher Institute for Environmental Sciences, Hong Kong Baptist University, Hong Kong SAR, China

Correspondence: ckcwong@hkbu.edu.hk

ligand–receptor methodology to identify putative STC1-binding proteins/receptors in human cells, followed by surface plasmon resonance (SPR) analysis to characterize the binding and kinetic affinity.

## Results

In the TriCEPs-LRC assay, the ligand protein was hSTC1-Flag, the positive control was transferrin, and the negative control was glycine. LC–MS/MS analysis of the positive control, transferrin receptor protein 1 (TFR1) was present as in all three replicate experiments (Fig S1). In contrast, no proteins were captured in the negative control condition. Comparing captured proteins by the ligand hSTC1-Flag versus the negative control (glycine), three STC1-binding proteins were identified, including STC2, PLOD1 (procollagen-lysine and 2-oxoglutarate 5-dioxygenase 1), and MPRI (cation-independent mannose-6-phosphate receptor/IGF2R) (Fig 1A). In comparing the captured proteins by the ligands hSTC1-Flag and transferrin, the binding proteins STC2, PLOD1, and MPRI (IGF2R) were consistently identified in hSTC1-capturing experiments (Fig 1B). In addition, OSBL5 (oxysterol-binding protein-related protein 5) was detected. In the shortlisted STC1-binding proteins, MPRI, PLOD1, STC2, and OSBL5 were identified with 32, 10, 2, and 2 peptides, respectively, by LC–MS/MS analysis (Fig S2). MPRI (IGF2R) is the one with the highest number of identified peptide and the only plasma membrane receptor. MPRI (IGF2R) was used to further characterize the binding affinity with hSTC1 using Biacore SPR, a real-time, label-free method for detecting biomolecular interactions.

Fig 2A displays the binding of hSTC1-His to immobilized hIGF2R-His. hSTC1-His binds strongly to immobilized hIGF2R-His. After regeneration with Pierce gentle Ag/Ab elution buffer, all bound hSTC1-His was removed. Fig 2B shows the reproducibility of hSTC1-His binding. Injection of hSTC1-His was followed by a dissociation period and regeneration. Injections of three successive hSTC1-His produced superimposed binding curves, suggesting almost identical binding responses. For the analysis of hSTC1 kinetic affinity, serially diluted samples of hSTC1-His were injected (Fig 3A). The binding responses were consistently large (up to 1,000 RUs) with slow on-rates and very slow off-rates. As a comparison, hSTC2-His and hCREG-His affinity responses were measured (Fig 3B and C). hSTC2-His binding produced significantly smaller responses (up to 100 RUs) than hSTC1-His, with faster on-rates and off-rates. hCREG (a known ligand of MPRI/IGF2R) induced large binding responses (up to 900 RUS) with slow on-rates and very slow off-rates. The binding curves were fitted to a single site binding model for the three ligands, although data fit better (lower $\chi^2$ value) to a two-state binding model (Fig 3A–C). The estimated $k_a$, $k_d$, and $K_D$ (equilibrium dissociation constant) are listed in Table S1. hSTC1-His bound to the immobilized IGF2R-His with high capacity (%$R_{max}$ = 70–100% of binding capacity) and high affinity in the 10–20 nM range. This indicates that hSTC1-His was bound with high affinity to most of the immobilized hIGF2R-His proteins. The results are similar to those obtained for hCREG-His. By comparison, hSTC2-His bound to the immobilized hIGF2R-His with a much lower capacity (%R max = 2–4% of binding capacity), but with high affinity (1–2 nM). There is

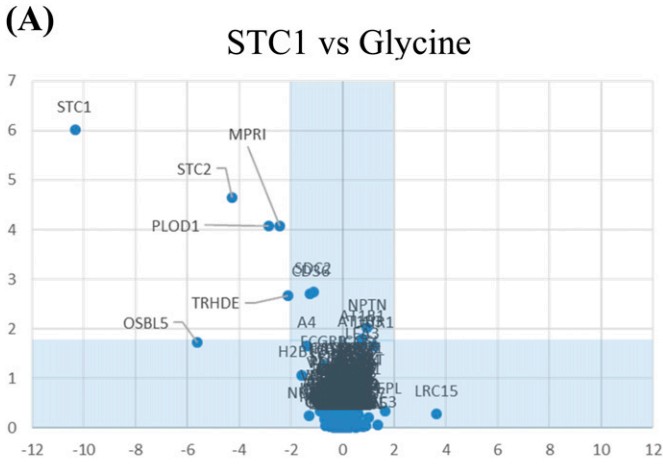

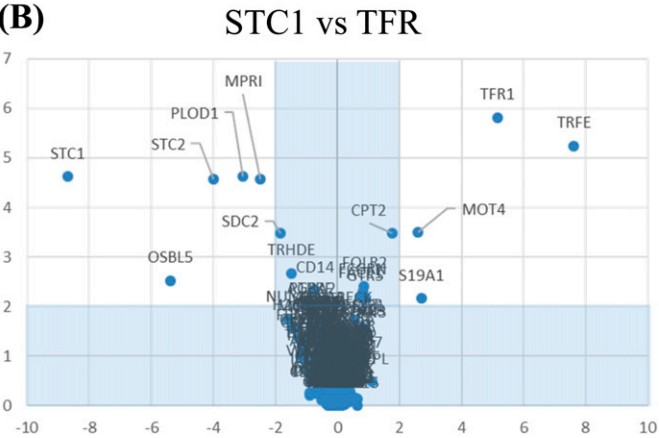

**Figure 1. TriCEPs-LRC assay.**
**(A, B)** Volcano plots of (A) hSTC1 and the negative control (glycine) and (B) hSTC1 and the positive control (transferrin) on captured protein candidates.

evidence that hSTC2-His bound with high affinity to a subpopulation of immobilized hIGF2R-His proteins, but did not bind to most of them.

To determine if hSTC1 competes with hCREG for binding to IGF2R, an SPR competitive assay was performed. hSTC1-His was immobilized in the CM5 sensor. The binding of hIGF2R-His to hSTC1-His produced responses of ~120 RUs (Fig S3). Fig 4 shows the results of the inhibition analysis. When hIGF2R-His was premixed with hCREG-His at increasing concentrations, the binding responses to hSTC1-His were significantly smaller than when hIGF2R-His was bound alone. The reduction of hIGF2R-His binding to hSTC1-His was dependent on the hCREG-His concentration. Initial binding on-rates were measured by fitting linear regression fits to binding responses from 2 to 10 s, which were 1.41, 0.93, 0.46, and 0.12 RUs/s for 0, 1, 2.5, and 5 µg/ml hCREG-His, respectively. Percent inhibition values were calculated as 0, 34, 68, and 91% for each hCREG concentration. hCREG-His alone, at increasing concentrations, did not bind hSTC1-His.

Our next step was to determine whether STC1 exerts its biological effects through IGF2R/MPRI. ThP-1 cells with or without IGF2R/MPRI knockdown were used to examine the effects of STC1 on IL-1β secretion. The silencing experiments showed significant reductions

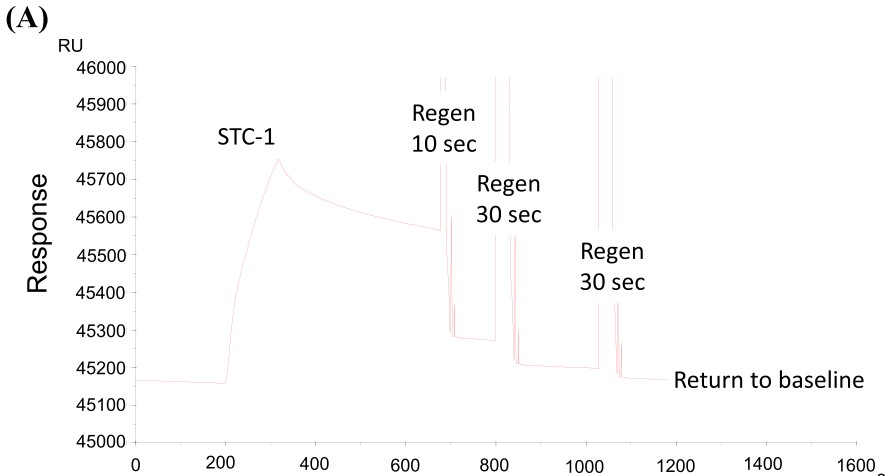

**(A)**

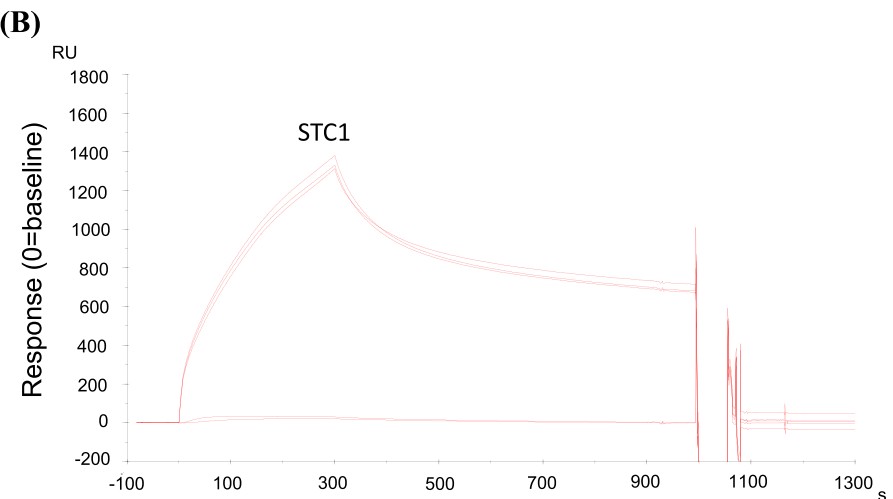

**(B)**

**Figure 2. Surface Plasmon Resonance Sensogram.**
**(A)** Binding of hSTC1 to immobilized IGF2R and regeneration of the CM5 sensor. hSTC1 was prepared at 5 μg/ml in the running buffer (PBS + 0.005% Tween 20) and was injected at 30 μl/minute and showed a large binding response. Following successive Pierce Gentle Ag/Ab Elution Buffer injections, all bound ligands were removed to baseline. **(B)** Reproducibility of hSTC1 binding. Three replicates of STC1 were prepared in the running buffer. The injection of hSTC1 was followed by dissociation and regeneration. A buffer blank (no binding response) was injected before and after hSTC1 binding cycles.

in IGF2R/MPRI mRNA and protein expression (Fig 5A). In siRNA$_{Ctrl}$-transfected cells treated with 0.5 μg/ml of STC1, IL-1β secretion was significantly decreased (Fig 5B). In siRNA$_{IGF2R/MPRI}$-transfection, the effect was not observed.

## Discussion

Using the LRC-TriCEPS method, we identify the membrane protein receptor, MPRI (also known as, IGF2R) as a potential receptor candidate. The receptor is a 270 kD polypeptide, widely expressed in many cell types. The receptor consists of a large extracellular domain, which can bind to multiple classes of ligands, including retinoic acid, IGF2, and CREG (cellular repressor of E1A-stimulated gene). Moreover, the receptor is essential for modifying non-lysosomal proteins, including proliferin, TGFβ precursor, epidermal growth factor receptor, leukemia inhibitory factor, and macrophage-colony-stimulating factor (Lee & Nathans, 1988; Purchio et al, 1988; Todderud & Carpenter, 1988; Blanchard et al, 1999). The receptor also plays a role in regulating extracellular levels of growth factors (i.e., transforming growth factor β1 and IGF2) and intracellular sorting of lysosomal enzymes. The importance of IGF2R/MPRI is known in the regulation of cell growth in embryonic development (Lau et al, 1994) and carcinogenesis (Martin-Kleiner & Gall, 2010). The loss of receptor function was associated with larger mutant mice (Wang et al, 1994) because it is the scavenger of IGF2. The binding of receptor to CERG inhibited the growth of cancer cells (Di & Gill, 2003). Intriguingly, embryonic studies have demonstrated the involvement of STC1 in fetal development and cell differentiation. A dwarf phenotype was observed in transgenic animals overexpressing STC1 (Johnston et al, 2010). Under oxidative stress, STC1$^{+/+}$ mouse embryonic fibroblasts grew significantly slower than STC1$^{-/-}$ cells (Nguyen et al, 2009). Several other growth-related functions of STC1 were demonstrated when the transcriptional program was studied in serum-stimulated human fibroblasts and the differentiation of neurons, megakaryocytoid cells, adipocytes, and osteoblasts (Yeung et al, 2012).

Using an SPR assay, the binding specificity of STC1 to IGF2R was assessed and quantified. The data confirm that STC1 binds to IGF2R/MPRI directly with a high affinity and capacity. The binding data are comparable with that between IGF2R/MPRI and CREG (a known ligand of the receptor). Comparatively, the paralog STC2 showed a much lower binding capacity than STC1. In SPR competitive binding assays, CREG competed with STC1 for binding to

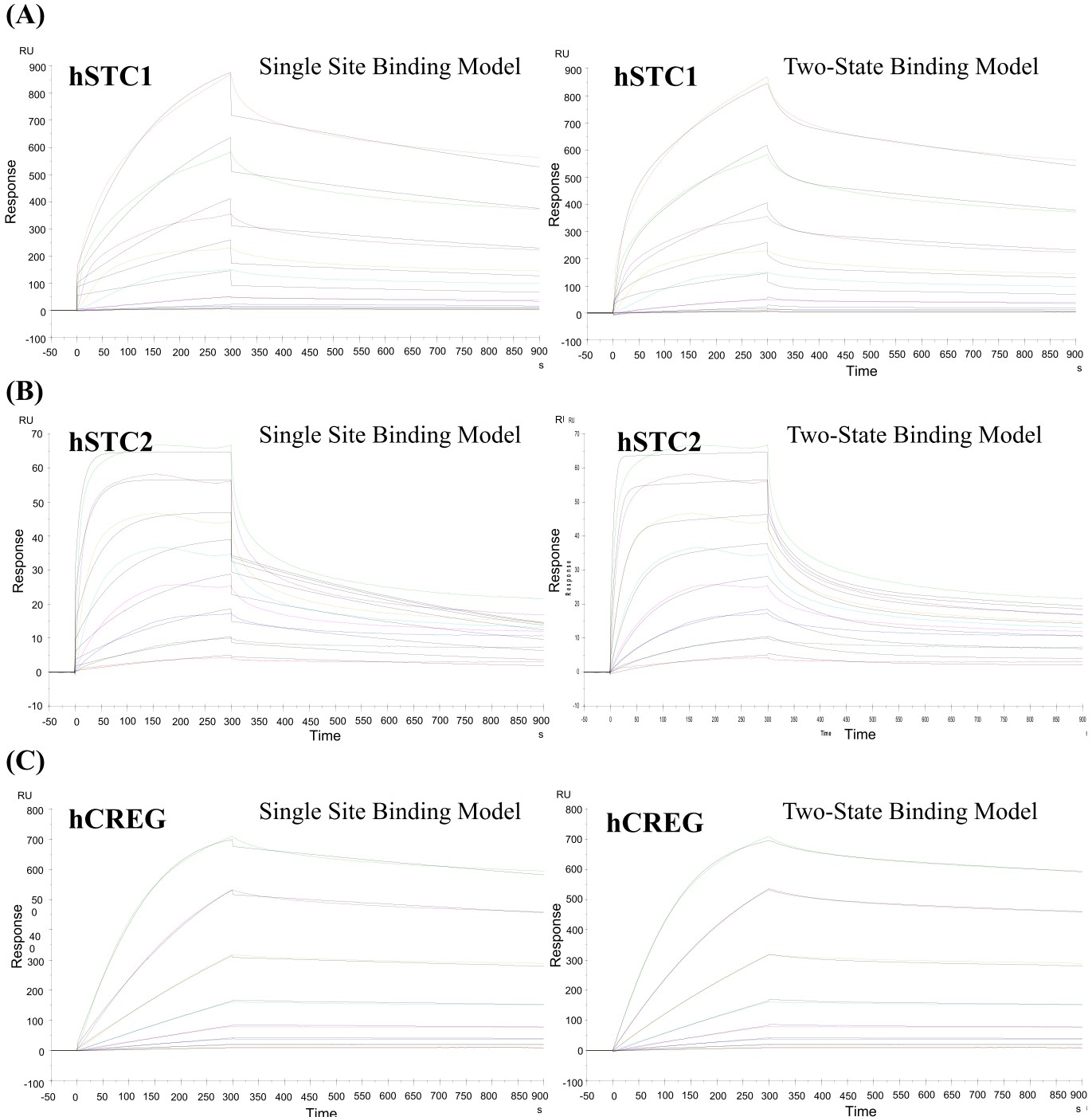

**Figure 3. Surface plasmon resonance sensorgrams.**
**(A)** hSTC1, **(B)** hSTC2, and **(C)** hCREG binding to human IGF2R.

IGF2R/MPRI. It has been reported that CREG binds to the extracellular domains 7–10 of the receptor in a glycosylation-dependent manner (Di & Gill, 2003). STC1 may also bind to the same receptor domains. Nonetheless, it is unclear how STC1 functions via IGF2R/MPRI.

The current literature supports STC1's role in inflammation. The expression of STC1 has been associated with the development of

hypoxic tumor microenvironment, which is similar to chronic inflammation (Chai et al, 2015). Tumor-associated macrophages, the main driver of tumor inflammation and progression, regulate the dynamics of cancer progression (Solinas et al, 2009). In macrophages, STC1 inhibits ROS by inducing mitochondrial uncoupling protein-2 (Wang et al, 2009). ROS are important upstream signals for NLRP3 (NOD-, LRR-, and pyrin domain-containing protein 3)

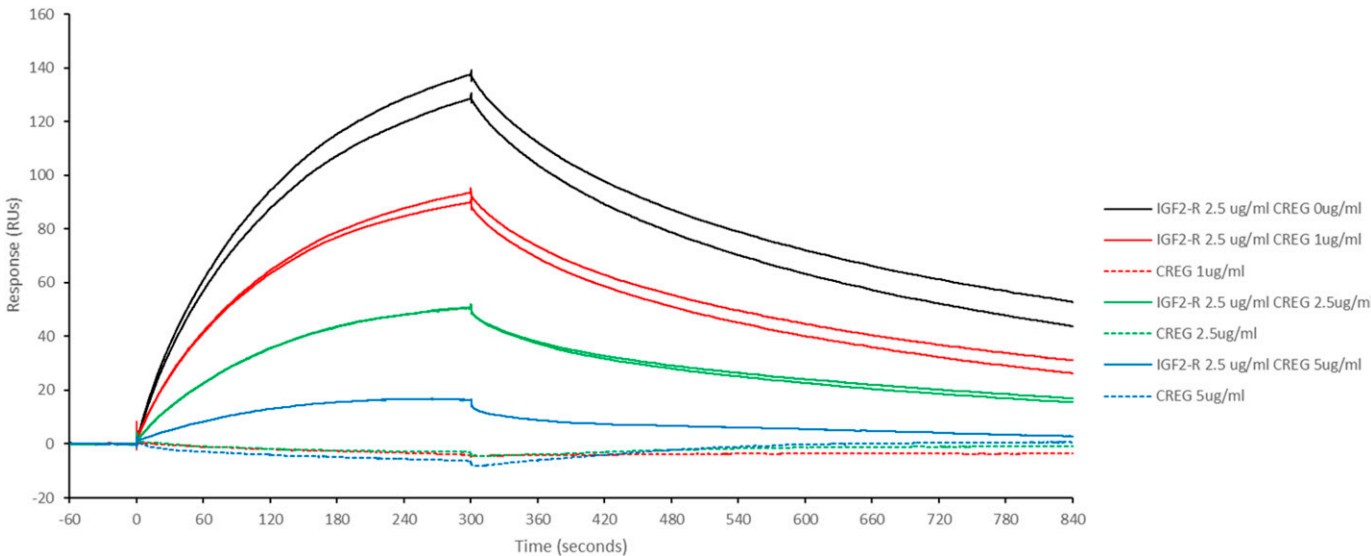

**Figure 4. Percentage inhibition analysis of IGF2R binding to hSTC1 by hCREG.**
Surface plasmon resonance sensorgram of immobilized hSTC1 binding to IGF2R, premixed with an increasing concentration (0 [black solid lines], 1 [red solid lines], 2.5 [green solid lines], and 5 µg/ml [blue solid lines]) of hCREG. IGF2-R alone binding to hSTC1 produced binding responses of ~120 RUs. IGF2R pre-mixed with hCREG at increasing concentrations gave smaller binding responses than IGF2R binding alone. Note that hCREG alone (dotted lines), at increasing concentrations, did not bind to hSTC1.

**(A)**

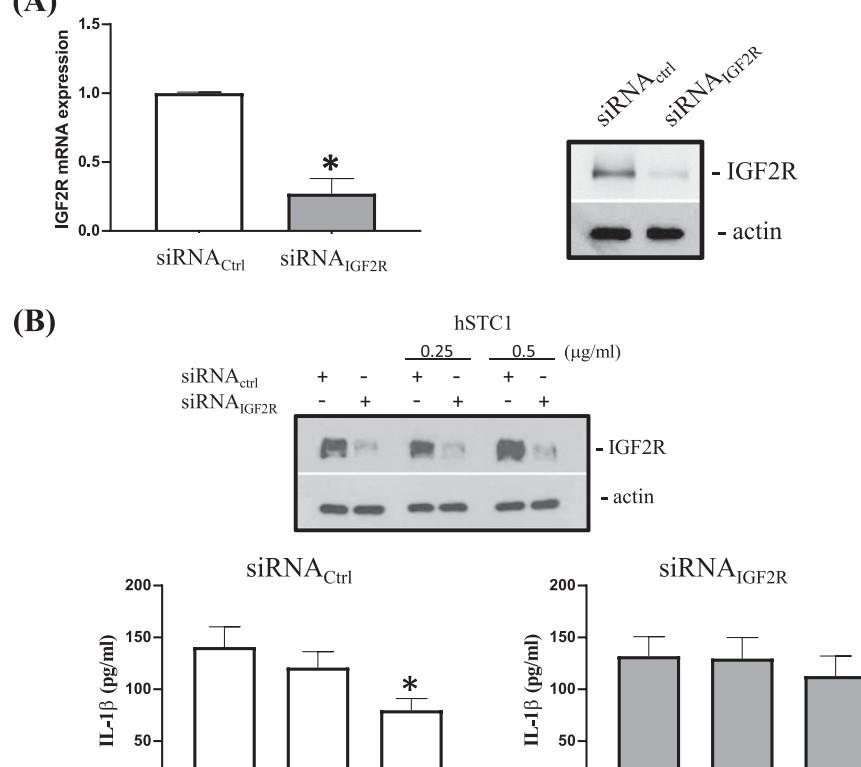

**Figure 5. IGF2R-silencing on STC1 inhibited IL-1β secretion.**
**(A)** The effect of siRNA-silencing on the mRNA and protein expression of IGF2R. **(B)** The effects of hSTC1 (0.25 and 0.5 µg/ml) on IL-1β secretion in siRNA$_{Ctrl}$ and siRNA$_{IGF2R}$-silencing cells. A significant reduction of secreted IL-1β was measured in hSTC1 treatment (0.5 µg/ml, *$P < 0.001$) as compared with the control. The inhibitory effect on IL-1β secretion was abolished in siRNA$_{IGF2R}$ transfection.
Source data are available for this figure.

**(B)**

inflammasome activation, so ROS blockade inhibits inflammasome activity. Previously, STC1 was shown to negatively regulate macrophage NLRP2 inflammasome activation to inhibit IL-1β secretion in human mesenchymal stem cell/macrophage co-culture (Oh et al, 2013). In retrospect, we studied STC1's effects on the secretion of the proinflammatory cytokine, IL-1β. In line with the previous finding, STC1 treatment reduced macrophage secretion of IL-1β via IGF2R/MPRI. Intriguingly, a recent study revealed that IGF2R-elicited anti-inflammatory phenotype in macrophages, accompanied with a significant reduction of IL-1β (Wang et al, 2020). The study supports our data to illustrate the role of IGF2R/MPRI in mediating the anti-inflammatory effects of hSTC1.

In summary, human STC1 is a newly identified ligand of the membrane receptor IGF2R/MPRI. It is possible that the receptor plays a role in the cellular sequestration of STC1 (Wagner & DiMattia, 2006). In addition, the results presented here show direct competitive receptor binding of CREG and hSTC1. Intriguingly, hSTC1, CREG, and IGF2R/MPRI are known to mediate cell growth, inflammation, and carcinogenesis. Therefore, further research on the binding interaction between the two ligands via IGF2R/MPRI is warranted.

# Materials and Methods

## Cell culture

Human embryonic kidney cell line, HEK293T, was cultivated in DMEM (Gibco) supplemented with 10% FBS and 0.5% penicillin/streptomycin. Human leukemia monocytic cell line, ThP-1 was maintained in RPMI 1640 medium (Gibco) containing 10% FBS and 0.5% antibiotics. The cell lines were all maintained at 37°C and 5% $CO_2$.

## Protein expression and purification

The human STC1 cDNA was subcloned into pDONR221 using Gateway Technology (Invitrogen) and then transferred to Gateway-compatible destination vectors with Flag-tag for expression studies. HEK293T cells were overexpressed with Flag-tagged human (h) STC1 using Lipofectamine-2000. After 24 h of transfection, the cells were washed and cultured in serum-free OPTI-MEM with a protease inhibitor cocktail (Roche). By using the anti-Flag (M2) antibody (Stratagene), we assessed whether hSTC1-Flag was present in the cell lysate and conditioned medium. The N-linked carbohydrate moieties of hSTC1-Flag were removed using peptide-N-glycosidase F (PNGase F) (New England BioLabs, Inc.). A protein sample was denatured by heating the reaction with 1X glycoprotein denaturing buffer at 100°C for 10 min, followed by incubation with 1X glycobuffer 2, 1% NP40, and PNGase F at 37°C for 1 h. This solution was subjected to Western blotting (Fig S4A). hSTC1-Flag containing media was collected twice every 48 h and centrifuged at 2,000g to remove dead cells. By using Amicon Ultra 10 K centrifugation units (Millipore), the supernatant was pooled and concentrated, and then purified by using M2 magnetic beads (Thermo Fisher Scientific). With 150 µg/ml 3xFLAG peptide (Thermo Fisher Scientific) in

PBS, hSTC1-Flag was eluted. As a reference to known concentrations of BSA, Coomassie blue staining was used to determine the amount of purified hSTC1-Flag (Fig S4B). The purified proteins were snap-frozen and stored at −80°C.

## Cyclic AMP assay

ThP-1 cells were incubated with 0.5 or 1 nM hSTC1-Flag in HBSS buffer containing IBMX for 45 min. Forskolin (FSK), a cAMP activator, was used as a positive stimulant. Cyclic AMP levels were measured using the Lance cAMP 384 kit (PerkinElmer Life Sciences). With the VICTOR X4 Multilabel Plate Reader (PerkinElmer Life Sciences), the fluorescent signal was measured at 340 nm and 615 and 665 nm (Fig S4C).

## Capturing of STC1-binding proteins

TriCEPS-based LRC and mass spectrometry were used to capture hSTC-1 targeted proteins on ThP1 cells. We coupled 20 µg of hSTC1-Flag with 10 µg of TriCEPS in a total volume of 50 µl Hepes (pH 8.2). The coupling reaction was then stopped and quenched with 20 µg glycine to prevent unspecific binding of TriCEPS to the cell surface during the subsequent binding study. As positive and negative controls for the experiment, transferrin (which targets the transferrin receptor) and glycine were coupled to TriCEPS, respectively. A dot blot with streptavidin-HRP was used to determine the efficiency of the coupling reaction (Fig S5A). In a flow cytometric analysis, ~7.5 × 10^5 cells were incubated with 1.2 µg of TriCEPS-coupled ligands in 100 µl of PBS (pH 6.5). For competition of TriCEPS-coupled hSTC1, the cells were pre-incubated with 10 times unlabeled STC1 before receptor capture. The cells were washed twice with 400 µl of ice-cold PBS buffer pH 6.5, labeled with streptavidin-R-PE, and analyzed by flow cytometry (Fig S5B).

As part of the MS-analysis of receptor-capturing, 300 µg of each ligand were coupled to 150 µg of TriCEPS in a total volume of 200 µl PBS. Cells were oxidized using 1.5 mM sodium metaperiodate at 4°C for 15 min, followed by washing and receptor-capturing at pH 6.5, 4°C for 90 min, using 2.5 × 10^7 cells per replicate. Streptavidin–agarose–resin solid-phase chromatography was used to purify target proteins after lysing the cells. Proteins were reduced, alkylated, and digested with trypsin after stringent washing. Thermo Orbitrap Elite spectrometer fitted with an electrospray ion source was used to analyze the tryptic peptides. Using a 15-cm C18 packed column, tryptic peptides were analyzed in data-dependent acquisition mode. The data were analyzed using Progenesis software. The following criteria were used to identify candidate proteins for interactions with TriCEPS-coupled ligand conjugates: (i) at least two peptides had fold changes > 4 and adjusted P-value < 0.01.

## Biacore SPR assay

His-tagged human insulin–like growth factor-2 receptor (hIGF2R-His, 6418-GR-050; R&D Systems) was prepared at 10 µg/ml in 10 mM sodium acetate buffer, at pH 5.5, 5.0, 4.5, and 4.0. To conduct pH scouting during immobilization, hIGF2R-His was injected for 1 min to determine optimum pH for isoelectric pre-focusing, where pH 5.0 showed the largest pre-focusing response and was chosen for the next step in immobilization (Fig S6A). The SPR

CM5 sensor chip was docked and primed with PBS for hIGF2R-His immobilization. The chip was then pre-cleaned with an injection of 50 mM NaOH. To activate carboxyl-dextran surfaces of flow cells, 1-ethyl-3-(3-dimethylaminopropyl)-carbodiimide (EDC, 400 mM) and N-hydroxy-succinimide (NHS, 100 mM) were mixed immediately before use. An injection of hIGF2R-His (10 µg/ml) in 10 mM sodium acetate buffer (pH 5.0) was followed by an injection of ethanolamine (1 M, pH 8.5) to deactivate/cap all unreacted EDC-NHS groups (Fig S6B). Human HIS-tagged STC1 (hSTC1-His, 9400-SO-050; R&D Systems) was prepared in PBS + 0.005% Tween 20 at a concentration of 10 µg/ml. For ligand binding, hSTC1-His (5 µg/ml) was injected into the running buffer for 5 min at 30 µl/min, followed by a 10 min dissociation period and a 1-min regeneration period.

To perform kinetic affinity assays, hSTC1-His, human His-tagged STC2 (hSTC2-His, 9405-SO-050; R&D Systems), and human His-tagged cellular repressor of E1A-stimulated gene (hCREG-His, 2380-CR-025/CF; R&D Systems) were prepared at 10 µg/ml in a running buffer, followed by a twofold serial dilution to give 10, 5, 2.5, 1.25, 0.625, 0.313, 0.156, 0.078, 0.039, and 0.019 µg/ml solutions. Data inspection and quality control were performed on all peptide dilution series. Analysis was omitted from curves with spurious artifacts, unstable, drifting responses, and slowly accumulating, non-saturable responses. In the next step, we fitted data to models of single-site kinetic affinity, two-state kinetic affinity, and equilibrium binding. To compare "goodness of fit" between single-site and two-state kinetic affinity models, $\chi^2$ values were examined. Theoretical $R_{max}$ values were calculated based on the level of protein immobilization, the molecular weight of the immobilized protein, and the molecular weight of the binding protein. The calculated $R_{max}$ values were derived from the fitted kinetic affinity curves.

### SPR competitive binding assay

A CM5 sensor was used for hSTC1-His immobilization, which was primed with PBS + 0.005% Tween 20, and pre-cleaned with 50 mM NaOH for 30 s. By injecting fresh EDC-NHS solution for 7 min, the sensor surface was activated. hSTC1-His prepared at 20 µg/ml in 10 mM sodium acetate buffer, pH 5.0, was injected for 7 min. A solution of ethanolamine (1 M, pH 8.5) was injected for 7 min to de-activate/cap all unreacted EDC-NHS groups. A total of 6,250 RUs of hSTC1-His were immobilized. A reference sensor was activated with EDC-NHS (without STC1), then capped with ethanolamine. hIGF2R-His binding and sensor regeneration were tested on immobilized hSTC1-His. hIGF2R-His (2.5 µg/ml) was injected for 5 min at 10 µl/minute in the running buffer (PBS + 0.005% Tween 20) to obtain a binding response. The sensor was then regenerated with successive injections of Pierce Gentle Ag/Ab Elution Buffer for 15 and 30 s to remove all bound receptors. In the competitive assay, hIGF2R-His was injected alone, or pre-mixed with hCREG-His (1, 2.5, or 10 µg/ml) in the running buffer to determine changes in the binding responses with immobilized hSTC1-His.

### Macrophage differentiation

ThP-1 cells were seeded at a density of $1 \times 10^5$ per ml and transfected using 50 nM siRNA$_{Ctrl}$ or siRNA$_{IGF2R}$ using siLenFect lipid reagent (Bio-Rad) in Opti-MEM (Invitrogen) for 24 h. The non-

targeting siRNAs (siRNA$_{Ctrl}$) UGGUUUACAUGUUGUGUGA, UGGUUUA-CAUGUUGUGUGA, and human IGF2R (siRNA$_{IGF2R}$) UCACAUGGGA-CACGGAAUA, AGACCAGGCUUGCU CUAUA were used in the gene silencing experiment. The cells were stimulated with 5 nM PMA for 6 h, followed by 5 nM PMA, 20 ng/ml IFNγ (Gibco, Life Technologies) and 100 ng/ml LPS (Invitrogen, Thermo Fisher Scientific) stimulation for another 18 h. To study effect of STC1, hSTC1-His (R&D Systems), with final concentration of 0.25 and 0.5 µg/ml were added immediately before the addition of IFNγ and LPS. Real-time PCR and Western blotting were used to validate the knockdown of IGF2R expression. Conditioned media were collected for ELISA for IL-1β.

### Real-time PCR and Western blot analysis

The cellular RNA was extracted using the TRIZOL Reagent (Gibco/BRL) according to the manufacturer's instructions. The $A_{260}/A_{280}$ of total RNA was >1.8, which was used to synthesize cDNA using the SuperScript VILO Master Mix (Invitrogen, Life Technologies). Real-time PCR was carried out with Applied Biosystems' Fast SYBR Green Master Mix. The primer sequences for human IGF2R: 5-GAGGGAAGAGGCAGGAAAG-3 and 5-TGTGGCAGGCATAC TCAG-3, human actin: 5-GACTACCTCATGAAGATCCTCACC-3, and 5-TCTCCTTA ATGTCACGCACGATT-3 were used. An mRNA level for human actin was used to normalize the data. In control amplifications, neither RNA nor reverse transcriptase was added.

Using cold radioimmunoprecipitation assay (RIPA) buffer (150 mM NaCl, 50 mM Tris–HCL, pH 7.4, 2 mM EDTA, 1% NP-40, and 0.1% SDS), cells were lysed and centrifuged at 12,000g for 10 min at 4°C. The protein concentration in the supernatant was determined using the DC Protein Assay Kit II (Bio-Rad) at 750 nm using a microplate reader (BioTek). The sample lysates were resolved in SDS–PAGE and transferred to PVDF membranes (Bio-Rad). After blocking with 5% non-fat milk in PBST for 1 h, the membrane was incubated with a primary antibody, rabbit anti-IGF2R (1:2,000, #20253-1-AP; Proteintech), or mouse anti-actin (1:10,000, #A2228; Sigma-Aldrich), followed by a secondary antibody, goat anti-rabbit, or horse anti-mouse (1:2,000) conjugated with HRP (Cell Signaling Technology). Chemi-Doc imaging system (Bio-Rad) was used to visualize protein bands using WESTSAVE Up (AbFrontier).

### ELISA

Conditioned media of the control and treated ThP-1 cells were collected, centrifuged at 13,523g, 5 min at 4°C, and kept at −80°C. IL-1β levels were determined using the human IL-1β/IL-1F2 DuoSet ELISA kit (DY201; R&D System). Briefly, the plate was coated with the capture antibody overnight at room temperature, followed by blocking with 1% BSA in PBS. Standard and samples were added, followed by incubation with the detection antibody. The streptavidin-HRP and substrate were added sequentially and incubated in dark. Using the SpectraMax absorbance microplate reader (Molecular Devices), the reading was measured at 450 and 540 nm.

### Statistical analysis

GraphPad Prism8 was used to analyze the data and presented it as mean ± SD using t test. A P-value of 0.05 indicates statistical significance.

# Supplementary Information

# Acknowledgements

This work was supported by the General Research Fund (HKBU12103817), Research Grant Council, Hong Kong.

## Author Contributions

HT Wan: investigation, methodology, and writing—original draft.
AHM Ng: data curation, formal analysis, investigation, and methodology.
WK Lee: investigation and methodology.
F Shi: investigation and methodology.
CK-C Wong: resources, formal analysis, supervision, funding acquisition, visualization, and writing—original draft.

## Conflict of Interest Statement

The authors declare that they have no conflict of interest.

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
