## [Reviewer comments · Life Science Alliance]

Life Science Alliance

Identification and Characterization of a Membrane Receptor that Binds to Human STC1

Hin Wan, Alice Ng, Wang Lee, Feng Shi, and Chris Wong

DOI: <https://doi.org/10.26508/lsa.202201497>

Corresponding author(s): Chris Wong, Hong Kong Baptist University

Review Timeline:

Submission Date:	2022-04-22
Editorial Decision:	2022-06-01
Revision Received:	2022-06-04
Editorial Decision:	2022-06-24
Revision Received:	2022-06-27
Editorial Decision:	2022-06-27
Revision Received:	2022-06-28
Accepted:	2022-06-28

Scientific Editor: Novella Guidi

Transaction Report:

June 1, 2022

Re: Life Science Alliance manuscript #LSA-2022-01497-T

Prof. Chris K Wong
Hong Kong Baptist University
Biology
Kowloon Tong
Hong Kong, Hong Kong HK
Hong Kong

Dear Dr. Wong,

Thank you for submitting your manuscript entitled "Identification and Characterization of a Membrane Receptor that Binds to Human STC1" to Life Science Alliance. The manuscript was assessed by expert reviewers, whose comments are appended to this letter. We invite you to submit a revised manuscript addressing the Reviewer comments.

Thank you for this interesting contribution to Life Science Alliance. We are looking forward to receiving your revised manuscript.

Sincerely,

B. MANUSCRIPT ORGANIZATION AND FORMATTING:

Reviewer #1 (Comments to the Authors (Required)):

This manuscript by Hin Ting Wan and colleagues used the TriCEPS-based ligand-receptor methodology to identify putative binding proteins for human STC1 (hSTC1) in human leukemia monocytic cell line, ThP-1. The LC-MS/MS analysis of peptides from shortlisted hSTC1-binding proteins detected STC2, PLOD1, and MPRI (IGF2R) in hSTC1-capturing experiments, in addition to OSBL5 (oxysterol-binding protein-related protein 5). In the shortlisted STC1-binding proteins, MPRI, PLOD1, STC2, and OSBL5 were identified with 32, 10, 2, and 2 peptides respectively by LC-MS/MS analysis. Because MPRI (IGF2R) had the highest number of identified peptide and the only plasma membrane receptor, the authors focused their analyses on STC1:MPRI, using Biacore surface plasmon resonance (SPR). The surface plasmon resonance (SPR) assay demonstrated that hSTC1 binds to immobilized IGF2R/MPRI with high affinity (10-20nM) and capacity (Rmax 70-100%). The receptor binding data are comparable with those of CREG (cellular repressor of E1A-stimulated gene) a known ligand of IGF2R/MPRI, with Rmax of 75 - 80% and affinity values of 1-2 nM. CREG competed with hSTC1 in binding to IGF2R/MPRI; however, it did not bind to hSTC1. Silencing IGF2R/MPRI significantly reduced IL-1 secretion in response to STC1 stimulation. The authors conclude that they identified for the first time the membrane receptor for hSTC1.

Major:

The authors present interesting data that warrant publication; however, this reader would like to bring to the authors' attention that another group identified megalin as a binding "receptor" for Stanniocalcin-1. Megalin binds STC1 and shuttles it from the cell-surface to the mitochondria (PMID: 29916093), and the binding occurs on conserved leucines in the signal peptide of megalin (PMID: 35046485). Thus, the statement that "we identified for the first time the membrane receptor for hSTC1" is incorrect.

As the authors acknowledged in the discussion, IGF2R is a multi-ligand receptor, and through trafficking to the lysosomes (from the cell-surface and Golgi), it serves to downregulate ligand function, including IGF2 and TGF- β 1. Other ligands known to bind IGF2R include retinoic acid, urokinase-type plasminogen activator receptor (UPAR, or PLAUR), and mannose-6-phosphate. Of note, many of the proteins that bind to IGF2R are mannose-6-phosphate-tagged. In this regard, is STC1 tagged with mannose-6-phosphate, and what happens to STC1 degradation when IGF2R is silenced? This would answer the question of whether IGF2R binding serves to channel STC1 for degradation or mediate its biological effects? Moreover, the choice of IL-6 as the biological assay is questionable. First, while the decrease in IL-6 after STC1 treatment in cells silenced for IGF2R is statistically significant, it was marginal. There are other known effects of STC1 that can be assayed, including the effects of STC1 on ROS generation, UCP2 expression, mitochondrial respiration, and glycolysis; why not examine these as biological assays?

Minor:

In the introduction section (page 3, line 6 from the bottom), the authors state that "In situ hybridization of rat kidneys identified mRNA expression of STC1 only in collecting ducts; however, proximal and distal tubules were highly immunoreactive to STC1 protein (31)." In situ hybridization studies by others (PMID: 24700878) revealed expression of STC1 mRNA throughout the kidney, including all tubule segments, blood vessels (including arterial walls) and cells within the glomeruli (tuft and parietal cells).

Reviewer #2 (Comments to the Authors (Required)):

Introduction

The introduction is well-written and gives a good summary of the history and function of STC-1 with a notable absence related to megalin shuttling of STC-1. Please update the manuscript based on this.

Megalin mediates plasma membrane to mitochondria cross-talk and regulates mitochondrial metabolism.

Li Q, Lei F, Tang Y, Pan JS, Tong Q, Sun Y, Sheikh-Hamad D. *Cell Mol Life Sci*. 2018 Nov;75(21):4021-4040. doi: 10.1007/s00018-018-2847-3. Epub 2018 Jun 9.

Interactions between leucines within the signal peptides of megalin and stanniocalcin-1 are crucial for regulation of mitochondrial metabolism.

Li Q, Holliday M, Pan JS, Tan L, Li J, Sheikh-Hamad D.

Results

The results are sound, clearly written, and well-controlled.

The results are typically presented in order based on figure number. I am not sure why the first figure cited is Supplementary Figure 2C.

Discussion

This is well-written and places the findings in proper context.

Reviewer #1:

Q1: The authors present interesting data that warrant publication; however, this reader would like to bring to the authors' attention that another group identified megalin as a binding "receptor" for Stanniocalcin-1. Megalin binds STC1 and shuttles it from the cell-surface to the mitochondria (PMID: 29916093), and the binding occurs on conserved leucines in the signal peptide of megalin (PMID: 35046485). Thus, the statement that "we identified for the first time the membrane receptor for hSTC1" is incorrect.

R1: Thank you for the information. We apologize for missing these two important papers. We have updated the references and corrected the statement.

Q2: As the authors acknowledged in the discussion, IGF2R is a multi-ligand receptor, and through trafficking to the lysosomes (from the cell-surface and Golgi), it serves to downregulate ligand function, including IGF2 and TGF- β 1. Other ligands known to bind IGF2R include retinoic acid, urokinase-type plasminogen activator receptor (UPAR, or PLAU), and mannose-6-phosphate. Of note, many of the proteins that bind to IGF2R are mannose-6-phosphate-tagged. In this regard, is STC1 tagged with mannose-6-phosphate, and what happens to STC1 degradation when IGF2R is silenced?

R2: We used STC1-His tagged from a commercial vendor (R&D) for the assay. The degradation process was not investigated in this manuscript, but it is on our next experimental agenda.

Q3: This would answer the question of whether IGF2R binding serves to channel STC1 for degradation or mediate its biological effects? Moreover, the choice of IL-6 as the biological assay is questionable. First, while the decrease in IL-6 after STC1 treatment in cells silenced for IGF2R is statistically significant, it was marginal. There are other known effects of STC1 that can be assayed, including the effects of STC1 on ROS generation, UCP2 expression, mitochondrial respiration, and glycolysis; why not examine these as biological assays?

R3: It is for this reason that we conducted a biological assay to determine whether STC1 elicits effects on ThP1 cells. In this study, we measured the effects of STC1 on secreted IL1- β (not IL6) in macrophages, which is in line with the known effects of STC1 on macrophages (Oh et. al., 2014, Stem Cells. 32:1553 -63, PMID: 24307525). In our case, we chose this assay because the biological assay is specific to macrophages and the functional phenotype is further downstream.

Q4: In the introduction section (page 3, line 6 from the bottom), the authors state that "In situ hybridization of rat kidneys identified mRNA expression of STC1 only in collecting ducts; however, proximal and distal tubules were highly immunoreactive to STC1 protein (31)." In situ hybridization studies by others (PMID: 24700878) revealed expression of STC1 mRNA throughout the kidney, including all tubule segments, blood vessels (including arterial walls) and cells within the glomeruli (tuft and parietal cells).

R4: You are correct. The manuscript has been revised accordingly.

Reviewer #2

Q1: The introduction is well-written and gives a good summary of the history and function of STC-1 with a notable absence related to megalin shuttling of STC-1. Please update the manuscript based on this.

-Megalín mediates plasma membrane to mitochondria cross-talk and regulates mitochondrial metabolism. Li Q, Lei F, Tang Y, Pan JS, Tong Q, Sun Y, Sheikh-Hamad D. Cell Mol Life Sci. 2018 Nov;75(21):4021-4040. doi: 10.1007/s00018-018-2847-3. Epub 2018 Jun 9.

-Interactions between leucines within the signal peptides of megalin and stanniocalcin-1 are crucial for regulation of mitochondrial metabolism.

Li Q, Holliday M, Pan JS, Tan L, Li J, Sheikh-Hamad D.

R1: Thank you for the information. We apologize for missing these two important documents. The manuscript has been updated.

Q2: The results are typically presented in order based on figure number. I am not sure why the first figure cited is Supplementary Figure 2C.

R2: I agree with you. The sections (Result and Materials & Methods) were arranged incorrectly. The order has been corrected.

June 24, 2022

Re: Life Science Alliance manuscript #LSA-2022-01497-TR

Prof. Chris K Wong
Hong Kong Baptist University
Biology
Kowloon Tong
Hong Kong, China

Dear Dr. Wong,

Thank you for submitting your revised manuscript entitled "Identification and Characterization of a Membrane Receptor that Binds to Human STC1" to Life Science Alliance. The manuscript has been seen by the original reviewers whose comments are appended below. While the reviewers continue to be overall positive about the work in terms of its suitability for Life Science Alliance, some important issues remain.

Our general policy is that papers are considered through only one revision cycle; however, given that the suggested changes are relatively minor, we are open to one additional short round of revision. Please note that I will expect to make a final decision without additional reviewer input upon re-submission.

Please submit the final revision within one month, along with a letter that includes a point by point response to the remaining reviewer comments.

To upload the revised version of your manuscript, please log in to your account: <https://lsa.msubmit.net/cgi-bin/main.plex>
You will be guided to complete the submission of your revised manuscript and to fill in all necessary information.

B. MANUSCRIPT ORGANIZATION AND FORMATTING:

Sincerely,

Reviewer #1 (Comments to the Authors (Required)):

The authors were partially responsive to critique. There is no denying that the authors have identified a receptor for STC1; their data are novel, insightful and warrant publication. However, in spite of the fact that both reviewers pointed out to the authors the

existence of two publications demonstrating binding of STC1 to megalin, the authors continue to claim that IGF2R is "the receptor" for STC1. There are a number of statements that need to be adjusted/corrected.

Line three in the abstract states that "Currently, there is no known receptor for this hormone." That is incorrect.

The last sentence in the abstract states that "In our study, we identified IGF2R/MPRI is the membrane receptor for hSTC1." That too is incorrect, as megalin is a membrane receptor.

The authors should exercise caution with their statements in light of the fact that their data demonstrate binding of STC1 to a number of "receptors" including PLOD1, STC2 and OSBL5 (in addition to MPRI/IGF2R). Moreover, IGF2R binds to STC2; yet, with different dynamics ["hSTC2-His binding produced significantly smaller responses (up to 100 RUs) than hSTC1-His"], indicating that this receptor is not specific to STC1.

This story is half-baked and the authors have been partially responsive to critique. By their own admission they state in the discussion that "The receptor can bind to multiple classes of ligands, including retinoic acid, IGF2, and CREG." "Moreover, the receptor is essential for modifying non-lysosomal proteins, including proliferin, TGF β precursor, epidermal growth factor receptor, leukemia inhibitory factor, and macrophagecolony-stimulating factor (Blanchard et al., 1999; Lee & Nathans, 1988; Purchio et al., 1988; Todderud & Carpenter, 1988). The receptor also plays a role in regulating extracellular levels of growth factors (i.e., transforming growth factor β 1, IGF2) and intracellular sorting of lysosomal enzymes."

In light of this knowledge, reason dictates that IGF2R similarly regulates the abundance of STC1 (and possibly STC2) at the cell surface by channeling it to the lysosomes for degradation, and a simple experiment would answer this question and provide meaningful insight into the significance of STC1 binding to IGF2R. As far as the IL-1b as a biological read, while the decrease in IL-1b after STC1 treatment in cells silenced for IGF2R is statistically significant, it was marginal, and likely not biologically significant.

Reviewer #1:

Q1: Line three in the abstract states that "Currently, there is no known receptor for this hormone." That is incorrect. The last sentence in the abstract states that "In our study, we identified IGF2R/MPRI is the membrane receptor for hSTC1." That too is incorrect, as megalin is a membrane receptor.

The authors should exercise caution with their statements in light of the fact that their data demonstrate binding of STC1 to a number of "receptors" including PLOD1, STC2 and OSBL5 (in addition to MPRI/IGF2R). Moreover, IGF2R binds to STC2; yet, with different dynamics ["hSTC2-His binding produced significantly smaller responses (up to 100 RUs) than hSTC1-His"], indicating that this receptor is not specific to STC1.

R1: Sorry for this. We have deleted and revised the sentences, respectively in the revised manuscript.

Q2: This story is half-baked and the authors have been partially responsive to critique. By their own admission they state in the discussion that "The receptor can bind to multiple classes of ligands, including retinoic acid, IGF2, and CREG." "Moreover, the receptor is essential for modifying non-lysosomal proteins, including proliferin, TGF β precursor, epidermal growth factor receptor, leukemia inhibitory factor, and macrophagecolony-stimulating factor (Blanchard et al., 1999; Lee & Nathans, 1988; Purchio et al., 1988; Todderud & Carpenter, 1988). The receptor also plays a role in regulating extracellular levels of growth factors (i.e., transforming growth factor β 1, IGF2) and intracellular sorting of lysosomal enzymes."

In light of this knowledge, reason dictates that IGF2R similarly regulates the abundance of STC1 (and possibly STC2) at the cell surface by channeling it to the lysosomes for degradation, and a simple experiment would answer this question and provide meaningful insight into the significance of STC1 binding to IGF2R. As far as the IL-1 β as a biological read, while the decrease in IL-1 β after STC1 treatment in cells silenced for IGF2R is statistically significant, it was marginal, and likely not biologically significant.

R2: We might not explain it clearly in the manuscript. We apologize for the confusion. The role of STC1 in reducing IL-1 β in macrophages has been demonstrated in an independent study (Oh et al., 2013). Additional two other independent studies have found that IGF2R-elicited anti-inflammatory phenotypes in macrophages, were accompanied by significant reductions in IL-1 β levels (Du et al. 2019; Wang et al., 2020). On the basis of their biological reads, we demonstrate the possibility that STC1 may be involved in the secretion of IL-1 β through IGF2R. In the revised manuscript, we address this issue in the "discussion section".

However, this manuscript's main focus is on identifying and characterizing the STC1 binding receptor.

Du L, Lin L, Li Q, Liu K, Huang Y, Wang X, Cao K, Chen X, Cao W, Li F, Shao C, Wang Y, Shi Y (2019) IGF-2 preprograms maturing macrophages to acquire oxidative phosphorylation-dependent anti-inflammatory properties. *Cell Metab.* 29, 1363–1375.e8

Oh JY, Ko JH, Lee HJ, Yu JM, Choi H, Kim MK, Wee WR, and Prockop DJ (2013) Mesenchymal stem/stromal cells inhibit the NLRP3 inflammasome by decreasing mitochondrial reactive oxygen species. *Stem Cells*. DOI: 10.1002/stem.1608

Wang X, Lin L, Lan B, Wang Y, Du L, Chen X, Li Q, Liu K, Hu M, Xue Y, Roberts AI, Shao C, Melino G, Shi Y, and Wang Y (2020) IGF2R-initiated proton rechanneling dictates an anti-inflammatory property in macrophages. *Sci Adv*, **6**. DOI: 10.1126/sciadv.abb7389

June 27, 2022

RE: Life Science Alliance Manuscript #LSA-2022-01497-TRR

Prof. Chris K Wong
Hong Kong Baptist University
Biology
Kowloon Tong
Hong Kong, China

Dear Dr. Wong,

Thank you for submitting your revised manuscript entitled "Identification and Characterization of a Membrane Receptor that Binds to Human STC1". We would be happy to publish your paper in Life Science Alliance pending final revisions necessary to meet our formatting guidelines.

- please upload your supplementary figures as single files
- please add the Twitter handle of your host institute/organization as well as your own or/and one of the authors in our system
- please use the [10 author names, et al.] format in your references (i.e. limit the author names to the first 10)
- please remove the panel A in your Figure 4 legend and in the figure callout in the main manuscript text; since this is the only panel in the figure, it does not need a panel designation
- please add your panel labels to Figure S3

A. FINAL FILES:

B. MANUSCRIPT ORGANIZATION AND FORMATTING:

**Submission of a paper that does not conform to Life Science Alliance guidelines will delay the acceptance of your

manuscript.**

The license to publish form must be signed before your manuscript can be sent to production. A link to the electronic license to publish form will be sent to the corresponding author only. Please take a moment to check your funder requirements.

Sincerely,

June 28, 2022

RE: Life Science Alliance Manuscript #LSA-2022-01497-TRRR

Prof. Chris K Wong
Hong Kong Baptist University
Biology
Kowloon Tong
Hong Kong, China

Dear Dr. Wong,

Thank you for submitting your Research Article entitled "Identification and Characterization of a Membrane Receptor that Binds to Human STC1". It is a pleasure to let you know that your manuscript is now accepted for publication in Life Science Alliance. Congratulations on this interesting work.

DISTRIBUTION OF MATERIALS:

Again, congratulations on a very nice paper. I hope you found the review process to be constructive and are pleased with how the manuscript was handled editorially. We look forward to future exciting submissions from your lab.

Sincerely,
